# Generation of ZSM-5 Nanocrystallites and Their Assembly into Hierarchical Architecture in a Phase-Transfer Synthesis

**Xiaoling Zhao** [1,2]**, Jinlong He** [1,2] **and Jinjin Li** [1,2,*]

1    Natural Gas Institute of PetroChina Southwest Oil & Gasfield Company, Chengdu 610000, China
2    National Energy R&D Center of High-Sulfur Gas Reservoir Exploitation, Chengdu 610000, China
*    Correspondence: lijinjin621@163.com; Tel.: +86-181-8093-7028

**Abstract:** A method of phase-transfer water/toluene synthesis was developed to fabricate mesocrystals of Zeolite Socony Mobil-5 (ZSM-5) that contain both meso-/micropores and nanometer crystallites. The construction of a hierarchical architecture from nanozeolites via oriented attachment growth was achieved by a simple phase-transfer water/toluene synthesis by minimizing classical atom-by-atom crystallization. This opens the way to the cheap, highly efficient engineering of zeolitic morphologies. The physicochemical properties of the crystal were revealed by powder X-ray diffraction (XRD), $N_2$ physical adsorption, inductively coupled plasma atomic emission spectroscopy (ICP-AES), scanning electron microscopy (SEM), transmission electron microscopy (TEM), ammonia temperature-programmed desorption ($NH_3$-TPD) and pyridine infrared spectroscopy (Py-IR), indicating that the material has a high specific surface area, mesopore volume and Lewis acid content. The hierarchical ZSM-5 exhibits a prolonged catalytic lifetime in dimethyl ether–methyl ether (DTO) conversion and enhanced selectivity for propylene owing to the enhanced structural properties. The method can be extended to the synthesis of other graded zeolites controlled by the crystallization process and produce crystals comprising traversing mesoporosity and ultrasmall crystallites that are crucial for mass transfer enhancement.

**Keywords:** phase-transfer water/toluene synthesis; ZSM-5; hierarchical architecture; dimethyl ether–methyl ether





## 1. Introduction

The formation of crystals involving the nucleation of nanocrystallites serving as building blocks, growth via oriented attachment of these building blocks, and the subsequent elimination of grain boundaries through Ostwald ripening has gained increasing attention as a means to generate hierarchical crystals with optimized architecture-dependent properties [1,2]. This nonclassical mechanism is regarded as the dominant crystallization pathway under certain conditions, for instance, the hydrothermal synthesis of microporous zeolites [3]. In particular, oriented attachment growth can be modified to generate zeolitic crystals with controlled morphological features, as have been observed for typical zeolites, such as Metal-substituted Aluminophosphates-5 (MeAPO-5) [4], Silicoaluminophosphate-11 (SAPO-11) [5], Zeolite Socony Mobil-5 (ZSM-5) [6], etc. Suppressing the contribution of concurrently occurring classical crystallization and obtaining hierarchical zeolite crystals in a one-pot synthesis remain formidable challenges.

Zeolites are a kind of inorganic crystal material with a microporous channel structure, and they are widely used in numerous industrial processes, including as acid catalysts, adsorbents, and ion exchangers. As solid acid catalysts, zeolites possess unique shape selectivity owing to their acidic properties, micropore size, topological structure, and morphology-dependent diffusion characteristics [7]. However, the solitary micropores of zeolite cause severe diffusion resistance, which seriously limits their further application [8]. This dilemma can be effectively circumvented by constructing hierarchical or

nanozeolites [9–11]. Hierarchical zeolites possess both auxiliary macro/mesopores in addition to their inherent micropore structure, attracting extensive attention compared with nanozeolites, which have the disadvantages of poor hydrothermal or catalytic stability, as well as technical difficulty in handling the processes of synthesis, shaping and upgrading. Many methods for constructing hierarchical zeolites have been proposed, such as hard templating, soft templating, posttreatment and so on [12,13]. Nevertheless, intertwined classical/nonclassical crystallization is often witnessed in conventional hydrothermal synthesis, making it difficult to rule out the contribution of classical atom-by-atom growth. Consequently, non-uniform zeolites with varied crystal sizes and shapes are produced. Moreover, it is often necessary to separate nucleation and aggregation so as to maintain mesoporosity between primary crystallites. To limit the classical crystallization contribution, several strategies have been proposed in recent publications, such as growth inhibitor modification, seeded growth, mechanochemical synthesis, microwaves, etc. For example, Rimer et al. introduced growth modifiers to charge into the synthetic gel of SSZ-13 (Stacey Zones (Chevron)) [14] and zeolite L [15], and the final crystal shape was mediated by the modifiers, resulting in the ability to control the crystal size, shape and morphology. Tang et al. synthesized anisotropic nanorod-bundle **MFI** (zeolite framework code according to the International Zeolite Association) mesocrystals through a salt-aided seed-induced organic-free method [16]. Normally, the crystals thus obtained do not contain mesopores but possess corrugated surfaces reflecting the crystallization history.

　　Alternatively, hierarchical zeolites can also be generated through crystallization kinetic control, with the advantage of cost-effective synthesis procedures. For example, Stucky et al. constructed single-crystalline core–shell hierarchical **MFI** crystals with tunable sizes via nucleation and the growth kinetic regulation without using a secondary template [17]. Rimer et al. introduced efficient accelerants of nucleation, such as polydiallyldimethylammonium (PDDA), into the synthetic gel of SSZ-13 to accelerate crystallization, resulting in the modification of crystal size [18]. Moreover, the concentration of the zeolite precursor was found to greatly affect the crystallization process, among which the concentrated zeolite precursor favored nucleation. Shi et al., for instance, used dry-gel conversion to construct a hierarchical structure of aluminosilicate aggregated by nanocrystals, which possessed both a relatively high surface area and high meso/micropore volumes [19]. In solvent-evaporation-induced self-assembly synthesis using modified dry-gel conversion (DGC), with the assistance of organosilane, hierarchical ZSM-5 [20] and ZSM-11 [21] can also be produced. Alternatively, as later shown for zeolite Beta [22] and SAPO-11 [5], when organic additives are charged into the synthetic gel after nucleation is conducted under DGC conditions but still before attached growth, it is possible to tailor the morphology to a high level of order. Indeed, house-of-card structures made of nanosheets and sponge-like hierarchical architectures have been accomplished in this way. However, two prominent drawbacks prevent the synthesis from becoming a practical method for large-scale production: 1. DGC is included in the nucleation process to increase the nucleation rate so as to control the crystallite size and the corresponding size distribution, which is troublesome for upscaling; 2. the crystallization process needs to be interrupted to charge the growth modifiers or porogens, making it a stepwise crystallization process. Actually, crystallization-process-controlled synthesis often requires special manipulations or setups, such as microwave heating, dry-gel preparation, or stepwise interrupted procedures, which could pose major obstacles for mass production. In order to circumvent the above barriers, Zhu et al. developed a novel phase-transfer synthesis route to prepare nanosized crystalline microporous materials in a water/toluene mixed solvent using common structure-directing agents. Several representative silicoaluminophosphate (SAPO) materials have been fabricated by the phase-transfer crystallization pathway, such as nanosized SAPO-31 [23,24] and SAPO-34 [25]. Unlike these protocols, in which all precursors reside in the same aqueous media and the separating interface is liquid/solid, in phase-transfer water/toluene synthesis, crystallites nucleated from the aqueous phase are extracted into the immiscible hydrophobic organic phase for further growth, which significantly reduces the size

of the final crystals due to higher supersaturation and, consequently, a faster nucleation rate. A question then arises as to whether this phase-transfer water/toluene synthesis can also be applied to generate the hierarchical architecture of aluminosilicate zeolites, the crystallization mechanism of which is presumed to occur via a unique mechanism.

It is attractive to synthesize hierarchical zeolitic crystals from tiny-nanozeolite assembly because the primary crystallites, as well as the pore network, can be controlled, which, on the other hand, often involves the initial stepwise preparation of preformed nanocrystallites and their controlled synthesis. Herein, we demonstrate that phase-transfer water/toluene synthesis can be tailored to generate hierarchically assembled ZSM-5, which is one of the most widely used catalysts in industry [26–28]. Such a synthesis not only enables us to retain the intrinsic acidity pertaining to zeolitic crystallinity but also offers a high level of control over the primary crystallite size and pore connectivity, resulting in good catalytic performance in converting dimethyl ether to olefin (DTO).

## 2. Results and Discussion

To explore the optimized composition range, the content of toluene, the Al source, the Si source and the Si/Al ratio in the synthesis system were systematically explored (Figures 1–9). Figure 1 shows that the amount of toluene added had no effect on the phase, and pure ZSM-5 could be obtained. The corresponding FE-SEM micrographs (Figure 2) confirmed that the morphological features varied with the toluene content in the phase-transfer synthesis, and all products had a relatively smooth crystal surface. Among them, ZSM-5-100-200 (Figure 2d–f) had the optimum morphological features with uniform spherical particles formed by stacking smaller nanocrystals. The corresponding textural properties were further obtained by measuring $N_2$ adsorption–desorption isotherms, as shown in Figures 3 and 10 and the corresponding calculated data are summarized in Table 1. Compared with other samples, ZSM-5-100-200 had a larger specific surface area and mesopore volume, which gradually decreased with increasing toluene content. Second, the Si/Al ratio of the initial synthetic gel was adjusted in the range of 50 to 150 while keeping the remaining components constant. The morphological features did not undergo significant changes at a lower Si/Al ratio, while they were markedly different when the Si/Al ratio was adjusted from 100 to 150 (Figure 5). At high Si/Al ratios, the synthesis method provided a bulky crystal with a smooth surface, attributed to the reduced inhibitory effect on crystal growth. Finally, the effect of the Al source on the phase and morphology was studied. Figure 6 shows the XRD patterns of ZSM-5 samples with Si/Al ratios of 50 and 100 synthesized by different Al sources in the toluene/water synthesis system. Only the typical **MFI** phase (JCPDS No. 44-0003) appeared to be free from impurities, proving that the type of Al source had no effect on the crystal phase structure. The corresponding FE-SEM micrographs are shown in Figure 7. All samples were composed of spheroidal particles with diameters of 150–280 nm, as observed in panoramic micrographs. The corresponding textural properties were further obtained by measuring $N_2$ adsorption–desorption, as shown in Figure 9, Figure 10 and Table 1. The results show that the type of Al source had little effect on the pore structure of the samples. The magnified images show that the morphology of the sample with a Si/Al ratio of 100 was relatively regular compared with that with a low Si/Al ratio, and all spherical particles were composed of smaller primary units. In addition, further experiments proved that the hierarchical structures cannot be acquired by non-monomer silicon (i.e., silica sol) as the silicon source (Figure 9). Collectively, extensive trials imply that an appropriate amount of toluene, Si/Al ratio and homogeneous nucleation are key to controlling the crystal morphology.

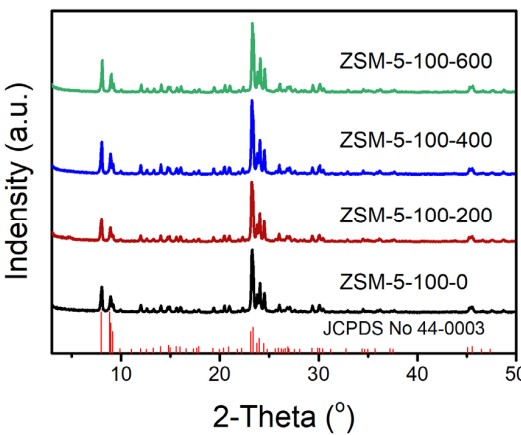

**Figure 1.** XRD patterns of ZSM-5 with Si/Al ratio of 100 synthesized with different amounts of toluene.

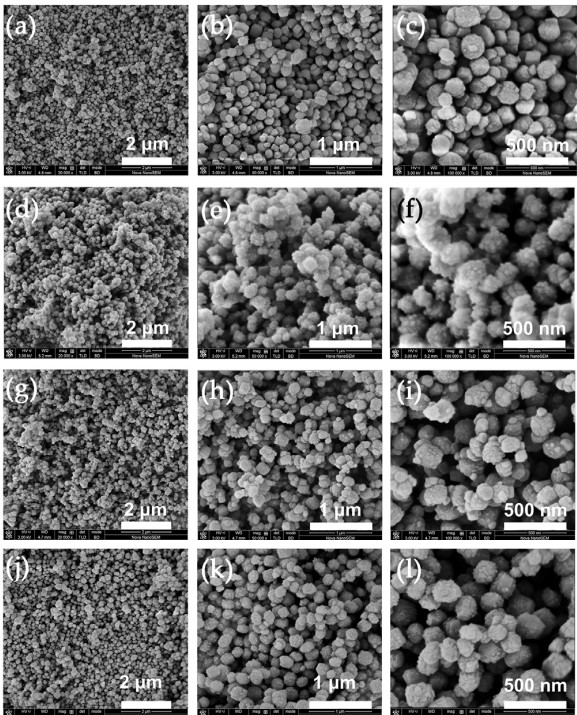

**Figure 2.** FE-SEM micrographs of ZSM-5 with Si/Al ratio of 100 synthesized with different amounts of toluene: (**a–c**): 0; (**d–f**): 200; (**g–i**): 400; (**j–l**): 600.

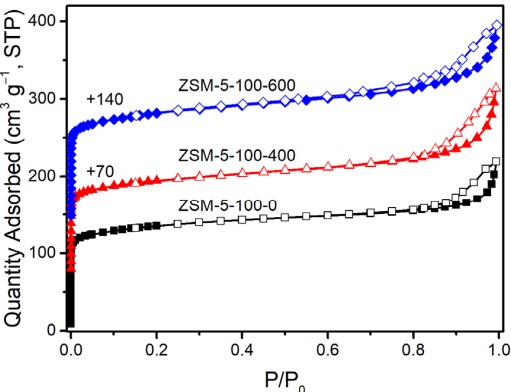

**Figure 3.** N$_2$ physisorption isotherms of ZSM-5 with Si/Al ratio of 100 synthesized with different amounts of toluene.

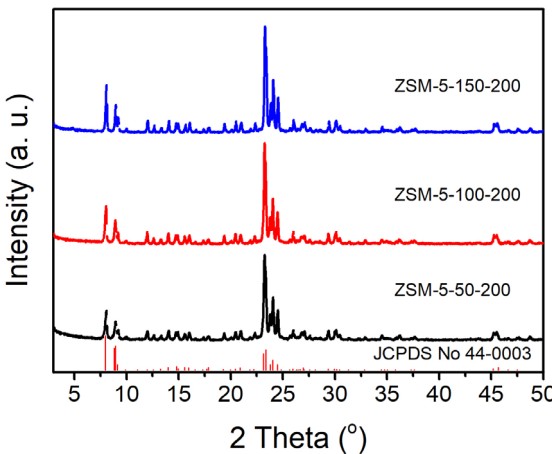

**Figure 4.** XRD patterns of ZSM-5 with different Si/Al ratios.

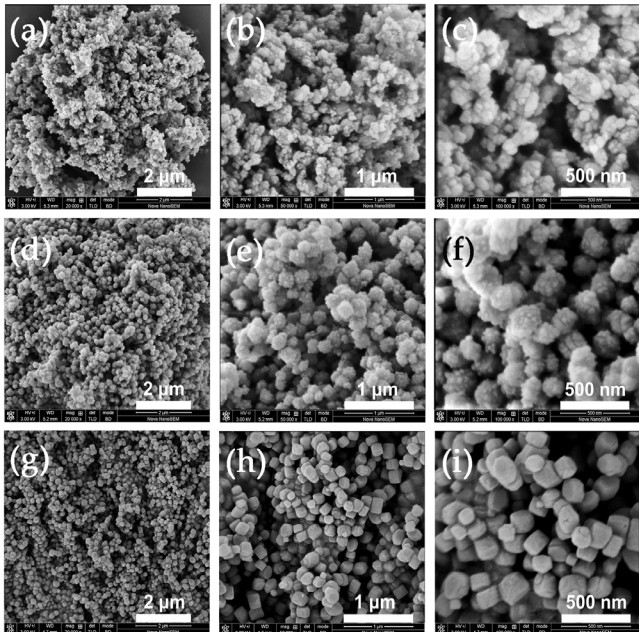

**Figure 5.** FE-SEM micrographs of ZSM-5 with different Si/Al ratios: (**a–c**): 50; (**d–f**): 100; (**g–i**): 150.

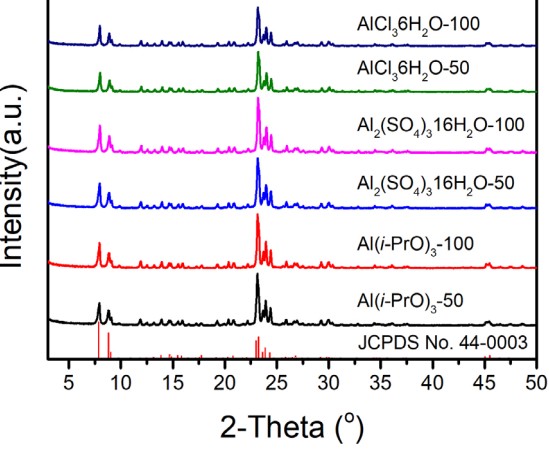

**Figure 6.** XRD patterns of ZSM-5 with Si/Al ratios of 50 and 100 synthesized with different Al sources.

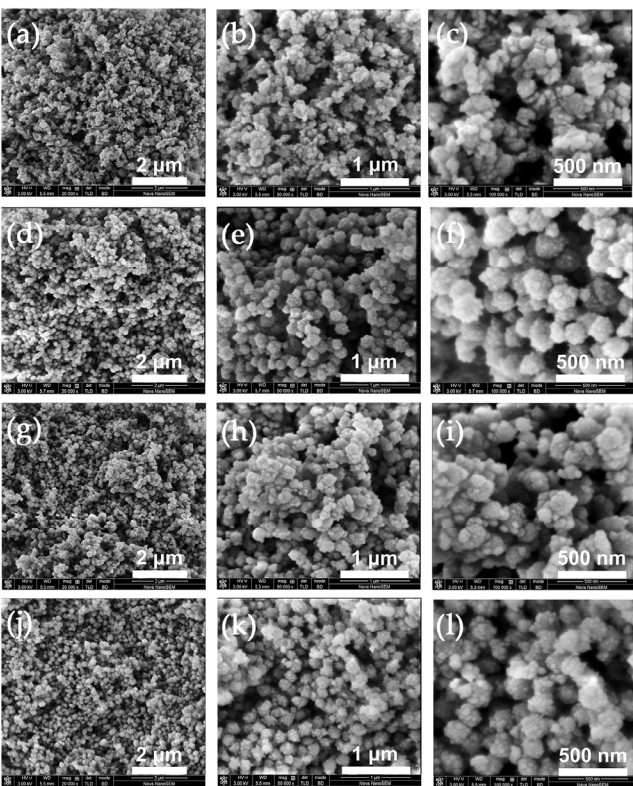

**Figure 7.** FE-SEM micrographs of ZSM-5 with Si/Al ratios of 50 and 100 synthesized with different Al sources. ZSM-5-50-200 (**a–c**) and ZSM-5-100-200 (**d–f**) synthesized with AlCl$_3$·6H$_2$O; ZSM-5-50-200 (**g–i**) and ZSM-5-100-200 (**j–l**) synthesized with Al$_2$(SO$_4$)$_3$·16H$_2$O.

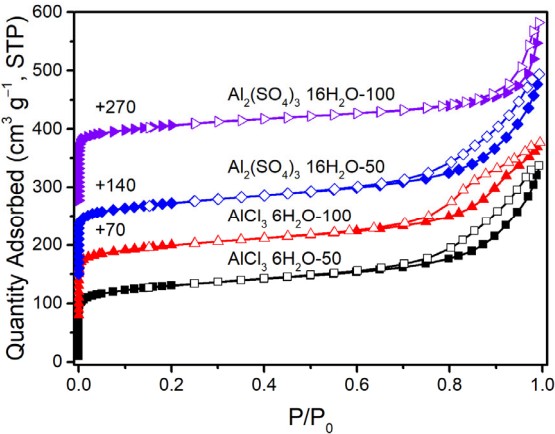

**Figure 8.** N$_2$ physisorption isotherms of ZSM-5 with Si/Al ratios of 50 and 100 synthesized with different Al sources.

Next, the systematized physicochemical properties of two representative samples, ZSM-5-H-50-200 and ZSM-5-H-100-200 synthesized with Al($i$-PrO)$_3$, were further characterized. TEM micrographs (Figure 11) of ZSM-5-100-200 indicated that spheroidal particles directionally accumulated in molten clusters with a size of about 33 nm in width and 70 nm in length. In addition, the black and white contrast indicated the presence of intercrystalline meso- or macropores. Directional accumulation alignment and growth are often an indicator of the history of directional attachment growth [29]. The textural properties of ZSM-5 samples were further obtained by measuring N$_2$ adsorption–desorption isotherms, as shown in Figure 10, and the corresponding calculated data are summarized in Table 1. When the relative pressure P/P$_0$ < 10$^{-3}$, the uptake curve of both samples exhibited a very steep rise owing to micropore filling. In addition, there was another jump at

$P/P_0 > 0.85$ accompanying a hysteresis loop in the isotherm, which was attributed to the capillary condensation of $N_2$ molecules in the interlayer mesopores or macropores formed by the accumulation of primary particles. The isotherms had both type I and type IV characteristics, indicating that the material contained both micropores and secondary meso- or macropores simultaneously. The surface area was 508 $m^2\ g^{-1}$ and 495 $m^2\ g^{-1}$ for ZSM-5-50-200 and ZSM-5-100-200, respectively. In addition, the distributions of pore sizes also showed the existence of a large number of interlayers mesopores and macropores in ZSM-5 samples (Figure 11b). Compared with ordinary NaZSM-5 [30], ZSM-5 synthesized by phase transfer had a larger specific surface area, mainly accounting for the increase in the external surface area and mesopore volume, which is consistent with the SEM and TEM results.

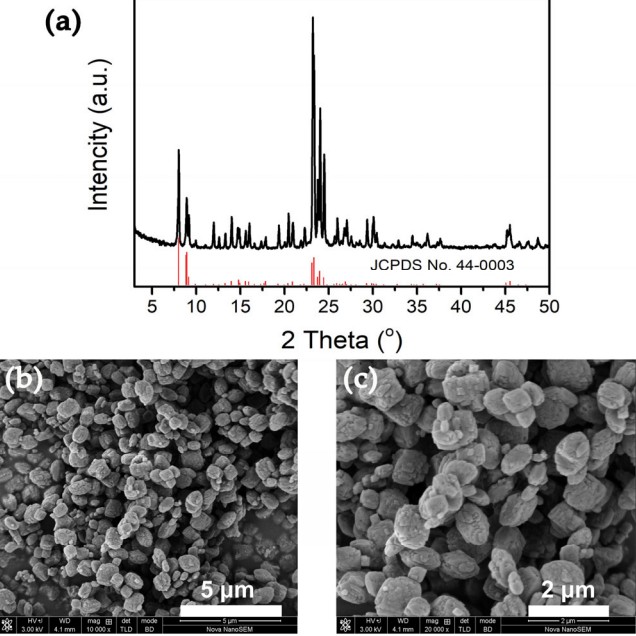

**Figure 9.** XRD patterns (**a**) and FE-SEM micrographs (**b**,**c**) of ZSM-5-100-200 synthesized with silica sol.

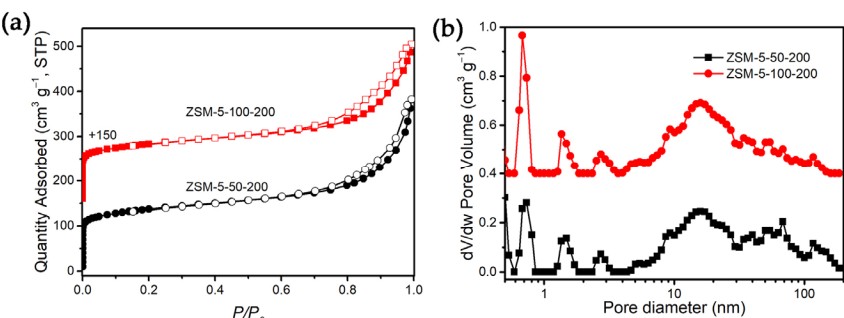

**Figure 10.** $N_2$ physisorption isotherms (**a**) and pore-size distribution obtained by the NLDFT method using the adsorption branch of the isotherm (**b**) for ZSM-5-50-200 and ZSM-5-100-200.

The acidic characteristics of the samples were measured by $NH_3$-TPD and Py-IR. From the $NH_3$-TPD profile (Figure 12), two maximum-temperature peaks of $NH_3$ desorption and two low-temperature peaks were observed for the two samples, corresponding to the desorption of $NH_3$ via chemical adsorption on the acidic site and physical adsorption on the surface Si-OH group, respectively [31]. In addition, the acidity of the sample decreased with the increasing Si/Al ratio due to the reduced negative charge of the inorganic framework. However, $NH_3$-TPD can only be used to qualitatively characterize the acidity of crystals, not quantitatively, and is also unable to distinguish the type of acid site. The acidity of the samples was further characterized by Py-IR. Vibration bands at around 1454 $cm^{-1}$ and

$1545 \text{ cm}^{-1}$ are attributable to pyridine adsorbed at Lewis acid sites and Brønsted acid sites, respectively, and the band at $1490 \text{ cm}^{-1}$ is ascribed to the joint influence of both sites [32,33]. The integrated areas of the given bands with their distinct molar extinction coefficients were used to calculate the amounts of adsorbed probe molecules at a given temperature ($\varepsilon_{Lewis}(1455 \text{ cm}^{-1}) = 2.22 \text{ cm } \mu\text{mol}^{-1}$ and $\varepsilon_{Brønsted}(1545 \text{ cm}^{-1}) = 1.67 \text{ cm } \mu\text{mol}^{-1}$) [34]. The results are provided in Figure 13 and Table 2. The Brønsted and Lewis acid site densities of the sample with a Si/Al ratio of 50 were relatively high, which is consistent with the $NH_3$-TPD result. In addition, the amount of Lewis acid markedly increased for hierarchical ZSM-5 zeolites compared with the ordinary samples [30], but the amount of Brønsted acid did not notably change, mainly due to the increase in the external surface area [35].

**Table 1.** Elemental composition and textural properties of ZSM-5-50-200 and ZSM-5-100-200.

| Sample | Si/Al [a] | $S_{BET}$ $(\text{m}^2 \text{ g}^{-1})$ [b] | $S_{ext}$ $(\text{m}^2 \text{ g}^{-1})$ [c] | $V_{total}$ $(\text{cm}^3 \text{ g}^{-1})$ [d] | $V_{micro}$ $(\text{cm}^3 \text{ g}^{-1})$ [c] | $V_{meso}$ $(\text{cm}^3 \text{ g}^{-1})$ [e] |
|---|---|---|---|---|---|---|
| ZSM-5-H-50-200 (Al(i-PrO)$_3$) | 45.6 | 508 | 132 | 0.59 | 0.16 | 0.43 |
| ZSM-5-H-100-200 (Al(i-PrO)$_3$) | 83.1 | 495 | 130 | 0.55 | 0.15 | 0.40 |
| ZSM-5-100-0 | \ | 465 | 41 | 0.34 | 0.16 | 0.18 |
| ZSM-5-100-400 | \ | 447 | 74 | 0.39 | 0.17 | 0.22 |
| ZSM-5-100-600 | \ | 470 | 69 | 0.38 | 0.17 | 0.21 |
| ZSM-5-H-50-200 (AlCl$_3$·6H$_2$O) | \ | 488 | 113 | 0.52 | 0.15 | 0.37 |
| ZSM-5-H-100-200 (AlCl$_3$·6H$_2$O) | \ | 482 | 121 | 0.47 | 0.15 | 0.32 |
| ZSM-5-H-50-200 (Al$_2$(SO$_4$)$_3$·16H$_2$O) | \ | 492 | 130 | 0.55 | 0.15 | 0.40 |
| ZSM-5-H-100-200 (Al$_2$(SO$_4$)$_3$·16H$_2$O) | \ | 511 | 103 | 0.48 | 0.18 | 0.30 |

[a] Determined by ICP-AES; [b] calculated using the BET method; [c] deduced from the *t*-plot method; [d] inferred from volume absorbed at $p/p_0 = 0.99$; [e] calculated from BJH method.

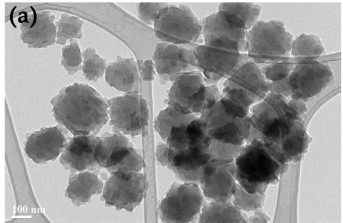
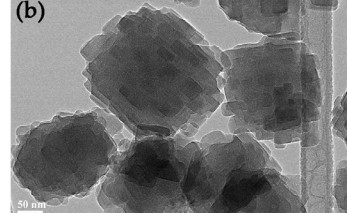

**Figure 11.** The panoramic (**a**) and enlarged TEM image (**b**) of ZSM-5-100-200.

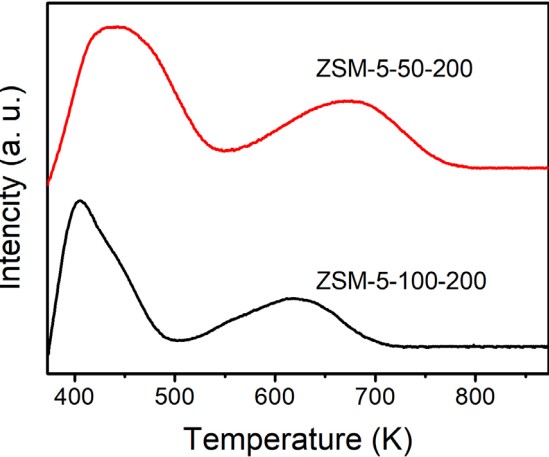

**Figure 12.** $NH_3$-TPD profile of ZSM-5-50-200 and ZSM-5-100-200.

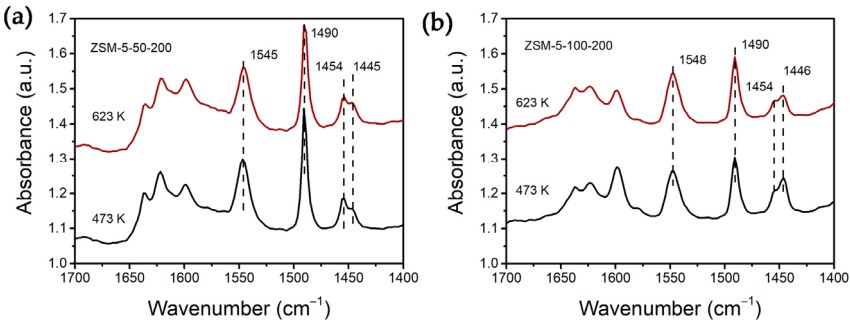

**Figure 13.** Py-IR profile of ZSM-5-50-200 (**a**) and ZSM-5-100-200 (**b**).

**Table 2.** Py-IR measurements of acidity for ZSM-5-50-200 and ZSM-100-200.

| Sample | Acidity Amount ($\mu$mol g$^{-1}$) | | | |
|---|---|---|---|---|
| | Brønsted Acidity (1545 cm$^{-1}$) | | Lewis Acidity (1455 cm$^{-1}$) | |
| | 473 K | 623 K | 473 K | 623 K |
| ZSM-5-50-200 | 240 | 224 | 109 | 87 |
| ZSM-5-100-200 | 114 | 66 | 100 | 63 |

In order to reveal the crystallization mechanism, ZSM-5-100-200 with different crystallization times was detected by ex situ photographs, XRD and SEM to monitor the crystallization process (Figures 14 and 15). In the early stage, a clear, colorless and transparent heterogeneous liquid mixture was formed in the biphasic medium, as shown by digital photographs. An obvious stratification appeared after standing for 0.5 min. Then, the underlying aqueous phase became milky white after dynamic crystallization for 2 h, mainly because the crystal sunk into the aqueous phase under the action of gravity. The solid product was found to have good oil solubility in the process of centrifugation, indicating that some kind of organic molecule was adsorbed on the crystal surface, possibly toluene, via π-π conjugation. With the extension of the crystallization time, the solid products gradually increased, and there was almost no change after 48 h. The change in crystal morphology was observed by FE-SEM during crystallization (tabk 15). FE-SEM tracking photography revealed that the initial crystal was made of spherical particles of ca. 130 nm with a rough surface. Thereafter, the crystal grew from 130 nm to 200 nm with increasing surface roughness. After 48 h, there was no obvious morphological change in the crystal. According to the above experiments, the crystal growth mechanism is proposed as follows: In the process of synthesis, zeolitic precursors dissolve in the aqueous medium, in which nucleation takes place. The toluene phase disperses the aqueous phase into a discontinuous phase during the dynamic crystallization process, and a certain amount of toluene molecules adsorb on the crystal surface via π-π conjugation, restricting the growth of the primary crystallite in the process of maturation growth. Further growth proceeds mainly through the oriented attachment mechanism to construct the final hierarchical architecture. Toluene and dynamic crystallization are the key factors in controlling the crystal morphology in the course of crystallization.

On the basis of the above experiments, the mechanism of crystallization in the toluene/water synthesis system is proposed to occur in the following three steps: nucleation, oriented attachment and the ripening growth process. In the early stage of crystallization, all of the components are dissolved in the aqueous phase. Under rapid tumbling and heating conditions, Si, Al and TPAOH react with each other to form amorphous nanocrystals. Then, they rapidly transform into crystal nuclei. Moreover, it is speculated that the supersaturation of the reaction system is high in the process of phase-transfer water/toluene synthesis, leading to fast nucleation and the consumption of large amounts of nutrients. As a result, the further growth of crystals in the later crystallization stage is inhibited to some extent. Secondly, subsequent nonclassical oriented aggregation growth occurs to form the final

uniform crystallites. In the ripening growth process, in situ hydrophobic surface modification by the adsorption of toluene can prevent their further growth.

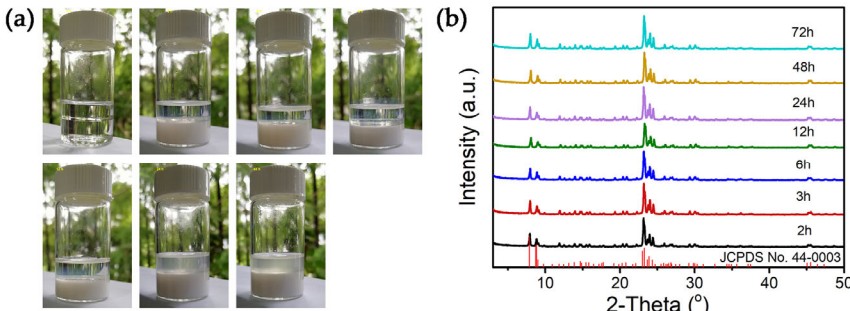

**Figure 14.** (**a**) Time-dependent digital photographs of ZSM-5-100-200 after standing for 0.5 min and (**b**) XRD patterns of the corresponding solid products collected at different time points in the course of the crystallization process.

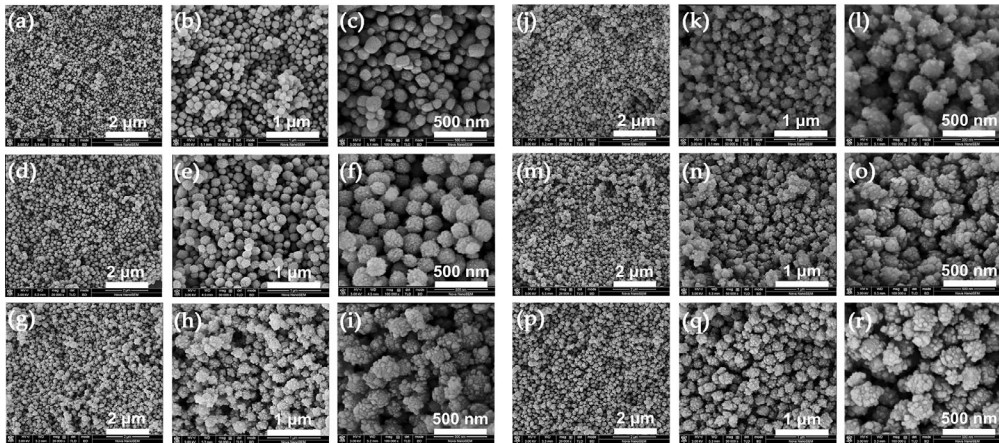

**Figure 15.** FE-SEM micrographs of ZSM-5-100-200 in the course of crystallization process: (**a–c**): 2 h; (**d–f**): 3 h; (**g–i**): 6 h; (**j–l**): 12 h; (**m–o**): 24 h; (**p–r**): 48 h.

The preliminary catalytic reaction performance for ZSM-5-50-200 and ZSM-5-100-200 was assessed for DTO and compared with ordinary ZSM-5 samples with the same Si/Al ratio, as shown in Figure 16 and Table 3 [30]. The curve of the conversion rate versus time on stream is shown in Figure 16a. A markedly prolonged catalytic life for hierarchal ZSM-5-50-200 (39 vs. 8 h) and ZSM-5-100-200 (83 vs. 10 h) was observed, which was mainly attributed to the improvement of the transport property due to the introduction of mesopores. In addition, the selectivity of propylene (ZSM-5-50-200: 49.2 vs. 43.1%; ZSM-5-100-200: *53.6* vs. 47.5%) was markedly improved compared with the control samples, as shown in Figure 16b. Moreover, the introduction of the hierarchical structure significantly reduced ethylene selectivity (ZSM-5-50-200: 11.4 vs. 17.6%; ZSM-5-100-200: 6.5 vs. 13.2%) compared with ordinary microporous samples, which was also reflected by the change in the ethylene/propylene ratio. For the reaction mechanism, the DTO reaction includes two cycle reaction paths, namely, the olefin cycle and the aromatic cycle, in which the conversion efficiency of the former is comparatively high. In the reaction process, the aromatic cycle can produce both ethylene and propylene, while the olefin cycle can only generate propylene [36,37]. As a measure of the cyclic advantage, the introduction of mesopores can shorten the residence time, thus suppressing the aromatic cycle to a certain extent and resulting in the alteration of alkene selectivity. Propane is considered to be a hydride transfer product of propylene. Therefore, a higher ratio of propane/propylene ratio in the products reflects the effect of hydrogen transfer on selectivity. The higher the ratio, the more alkenes are converted to alkanes. In addition, butene is considered to be

a product or intermediate of the olefin cycle, and its selectivity increases simultaneously with propylene, further proving that the introduction of mesopores increases the selectivity of the olefin cycle. In addition, the higher the density of acid sites, the more conducive to aromatic cycling, which is also the reason for the higher ethylene selectivity of ZSM-5-50-200 (11.4%) compared with ZSM-5-100-200 (6.5%) [38]. In general, hierarchical ZSM-5 samples showed high propene selectivity and coking resistance, which are the two most important parameters in DTO reactions, mainly attributed to the enhancement of diffusion properties, which inhibit aromatic cycling to a certain extent.

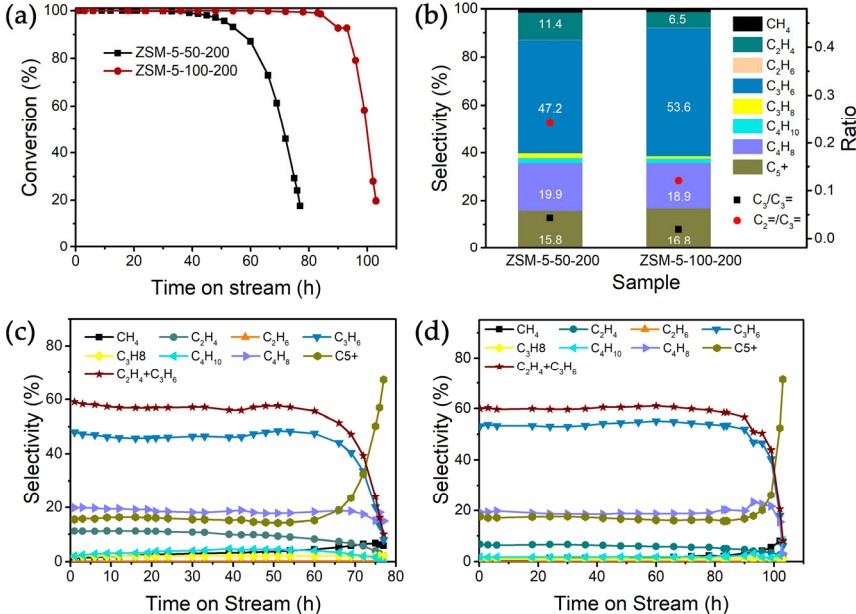

**Figure 16.** DME conversion (**a**) and overall product selectivity (**b**); selectivity for various products with TOS for ZSM-5-50-200 (**c**) and ZSM-5-100-200 (**d**).

**Table 3.** DTO catalytic result of ZSM-5-50-200 and ZSM-100-200 compared with ordinary ZSM-5 samples.

| Sample | Catalytic Life (h) | Selectivity (%) | | | | | | | | | |
|---|---|---|---|---|---|---|---|---|---|---|---|
| | | $CH_4$ | $C_2H_4$ | $C_2H_6$ | $C_3H_6$ | $C_3H_8$ | $C_4H_{10}$ | $C_4H_8$ | $C_5+$ | $C_3/C_3=$ | $C_2=/C_3=$ |
| ZSM-5-50-200 | 39 | 1.45 | 11.40 | 0.16 | 47.19 | 2.01 | 2.11 | 19.87 | 15.81 | 0.0426 | 0.2416 |
| ZSM-5-100-200 | 83 | 1.33 | 6.47 | 0.11 | 53.56 | 1.01 | 1.81 | 18.91 | 16.80 | 0.0188 | 0.1208 |
| ZSM-5-C-50 [a] | 8 | 2.78 | 17.6 | 0.21 | 43.10 | 5.61 | 17.39 | 0.48 | 12.83 | 0.1302 | 0.4084 |
| ZSM-5-C-100 [a] | 10 | 1.92 | 13.22 | 0.13 | 47.51 | 3.58 | 16.91 | 0.86 | 15.87 | 0.0754 | 0.2783 |

[a] The catalytic results of ZSM-5 C-50 and ZSM-5 C-100 y reprinted/adapted with permission from Ref [30]. Copyright 2021 by Wiley.

## 3. Materials and Methods

Tetraethyl orthosilicate (TEOS, 98%, Shanghai Aladdin Biochemical Technology Co., Ltd. Shanghai, China) and silica sol (40% in water, Shanghai Kaiyin Chemical Reagent Co., Ltd., Shanghai, China) were selected as the silicon source; aluminum isopropoxide (Al(*i*-PrO)$_3$, 98%, TCI), aluminum sulfate hexadechydrate (Al$_2$(SO$_4$)$_3$·16H$_2$O, 98%, Shanghai Aladdin Biochemical Technology Co., Ltd., Shanghai, China) and aluminum chloride hexahydrate (AlCl$_3$·6H$_2$O, 98%, Shanghai Aladdin Biochemical Technology Co., Ltd., Shanghai, China) were used as the aluminum source; tetrapropylammonium hydroxide (TPAOH, 40.0 wt.% in water, Shanghai Titan Reagents Co., Ltd., Shanghai, China) was used as the structure-directing agent (SDA); and deionized water and toluene (99.8 wt.%, Shanghai Aladdin Biochemical Technology Co., Ltd., Shanghai, China) were used as the synthetic solvent. All synthetic raw materials were used without further purification.

Hierarchical ZSM-5 samples were synthesized in a toluene/water mixture according to an initial gel molar ratio of $xAl_2O_3{:}100SiO_2{:}36TPAOH{:}2000H_2O{:}y$ toluene (x = 0.3, 0.5 or 1; y = 0, 200, 400 or 600). In a typical synthetic batch, SDA, the aluminum source (Al(*i*-PrO)$_3$, Al$_2$(SO$_4$)$_3 \cdot 16$H$_2$O, AlCl$_3 \cdot 6$H$_2$O) and the silicon source were added to deionized water in turn. After vigorous stirring for a certain time, a colorless, transparent homogeneous solution was obtained and named solution A. Next, solution A and an amount of toluene were simultaneously added to a Teflon-lined stainless-steel autoclave to undergo crystallization in a drying oven at 423 K with a rotational speed of 60 rpm for 72 h. After crystallization, the autoclave was rapidly cooled to room temperature by rinsing with tap water. The white solid product was obtained by centrifugation at 10,000 rpm for 10 min, followed by washing with deionized water to neutral and drying in a convection oven at 353 K for 8 h. Finally, the dried solid product was ground and placed in a muffle oven to calcinate at 823 K for 8 h in an atmosphere of air to remove residual SDA. The synthesized solid product is denoted as ZSM-5-x-y, where x and y represent the Si/Al ratio of the initial gel and the added amount of toluene, respectively.

In order to track the crystallization process of ZSM-5-100-200, crystallization was carried out for different times and quenched with tap water to obtain a series of solid products. The same process as above was used to treat the samples. In addition, the reaction emulsion was collected for further analysis.

The structure of the powder crystal was examined by X-ray diffraction (XRD) with Cu K$\alpha$ = 1.5418 Å for the X-ray source, operating at 40 kV and 100 mA on a Rigaku D/Max 2550 VB/PC diffraction, with a sweep range of 3–50° and scanning speed of 10° min$^{-1}$. The morphology of the crystals was photographed using field-emission scanning electron microscopy (FE-SEM; NOVA Nano SEM 450, Palo Alto, CA, USA). Transmission electron microscopy (TEM) images were photographed on a JEM-2011 (JEOL, Akishima-shi, Tokyo, Japan) operating at 200 kV. N$_2$ physisorption was performed at 77 K with an ASAP 2020 (Micromeritics, Atlanta, GA, USA) instrument. Prior to measurement, all samples were gassed under vacuum at 623 K for 12 h to remove contaminants adsorbed on the surface. The surface area of the samples was determined by the Brunauer–Emmett–Teller (BET) method. The non-local density functional theory (NLDFT) method was used to determine the pore volume of samples. Inductively coupled plasma atomic emission spectrometry (ICP-AES) (Agilent Technologies, Palo Alto, CA, USA) was used to measure the elemental composition of the crystals on an IRIS 1000 instrument. The NH$_3$ temperature-programmed desorption (NH$_3$-TPD) curve of samples was collected on a Chemisorb 2720 analyzer (Micromeritics Co., Atlanta, GA., USA). Pyridine desorption infrared spectroscopy (Py-IR) of the samples was performed on a Fourier transform infrared spectrometer (Nicolet Co., Madison, WI, USA) to analyze the acidity of ZSM-5 samples.

The catalytic performance of ZSM-5 samples was evaluated by the DTO reaction on a fixed-bed reactor at 673 K with a weight hourly space velocity (WHSV) of 4 h$^{-1}$ g$_{\text{-DME}}$ g$_{\text{-cat}}^{-1}$. The catalyst was first pretreated in an air stream at 823 K for 3 h before testing. The catalyst was then cooled to the reaction temperature (723 K) under a flow of N$_2$. Then, the reaction gas, dimethyl ether, was introduced into the reactor by N$_2$ and reacted. The gas composition at the reaction outlet was analyzed using an online gas chromatograph (GC) equipped with a flame detector (FID) and a Plot-Q chromatographic column (Agilent J&W GC chromatographic column, HP-PLOT/Q 19091P-Q04, 30 m × 320 m × 20 m) during the process of the reaction. Among them, methanol and dimethyl ether (DME) were considered reactants, and efflux hydrocarbons (HC) were considered products.

## 4. Conclusions

The phase-transfer water/toluene synthesis of hierarchical ZSM-5 zeolites was achieved in a single-step hydrothermal synthesis under agitation, hence paving a general avenue for the fabrication of hierarchical zeolitic materials. The synthesis procedure is simple and consists of conventional hydrothermal conditions as well as readily available starting materials. The synthesis enables high-order control over the crystallite size and pore network, which have

been proven to be key parameters that determine mass transfer properties. For example, the synthesized hierarchical ZSM-5 samples had a larger specific surface area, mesopore volume and Lewis acid site density, which resulted in a prolonged catalytic lifetime in DTO conversion and enhanced selectivity for propylene. Moreover, an underlying three-step mechanism was revealed for their formation: fast nucleation to produce uniform crystallites, in situ hydrophobic surface modification by the adsorption of toluene molecules that inhibit their further growth via the atom-by-atom route and the subsequent oriented aggregation growth pathway. This insightful knowledge shows that oriented attachment growth, as a nonclassical crystallization pathway, can be utilized to tune the architecture of zeolitic materials while simultaneously maintaining the intrinsic advantages of the crystalline zeolite framework, i.e., acidity and stability. The extrapolation of this strategy to other zeolitic materials and the fine-tuning of the architecture and composition for specific catalytic use are currently underway, and the new findings will be reported in future publications. The synthesis, crystallization pathway and materials are of interest from both academic and application perspectives.

**Author Contributions:** Formal analysis, data curation, writing—original draft preparation and writing—review and editing, X.Z.; supervision, J.H.; project administration, J.L. All authors have read and agreed to the published version of the manuscript.

**Funding:** This research was funded by the Postdoctoral Research Fund of PetroChina Southwest Oil & Gasfield Company, grant number [20220306-10].

**Data Availability Statement:** Not applicable.

**Acknowledgments:** This work was supported by the Natural Gas Institute of PetroChina Southwest Oil & Gasfield Company. The authors would like to express their sincere thanks to the Natural Gas Purification Institute of Natural Gas Institute of PetroChina Southwest Oil & Gasfield Company for its support of this work.

**Conflicts of Interest:** The authors declare no conflict of interest.

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
