# Peer review of "Generation of ZSM-5 Nanocrystallites and Their Assembly into Hierarchical Architecture in a Phase-Transfer Synthesis"

_catalysts, doi:10.3390/catal12101216_

Round 1
Reviewer 1 Report
-Should define all abbreviation in abstract, introduction,... as(as MeAPO-5,[4] SAPO-11,[5] ZSM-5,[6]).
-Where is the error bar in Fig 14.
-Compare the activity results with previous studies in table.
-Compare the activity results with previous studies in table
Reviewer 2 Report
Manuscript by Zhao entitled „Generation of ZSM-5 nanocrystallites and their assembly into Hierarchical architecture in a phase-transfer synthesis“ reports on synthesis of ZSM-5 zeolites in the water/toluene liquid mixture resulting in materials with improved catalytic properties in DTO conversion. Modification of structure of acidic zeolites in order to obtain more stable catalytic system, especially by simple and easily scalable method, is very important and interesting for readers.
Although the manuscript is interesting, I have several comments regarding the presentation of the data and their discussion:
- I'm a little lacking in textural data for materials synthesized at different water/toluene ratios, at different Si/Al ratios, using different Al sources, etc. I think it would adequately complement the overall picture of the prepared materials and the influence of individual variables on textural properties.
- In the experimental part, the NL-DFT method of evaluating adsorption isotherms is mentioned - from it, information on the distribution of pore sizes can be obtained, which, however, is missing in the manuscript.
- Fig. 9 contains adsorption isotherms from which the authors discuss the existence of mesopores and capillary condensation. It should be mentioned that the interpretation of such isotherms is not unambiguous, since the crystallites have dimensions of the order of 100-150 nm, they will have a relatively large external surface by themselves, and at the same time, for such small particles, condensation may occur in the interparticle space, which can be mistakenly interpreted as structural mesopores of crystallites. The data presented in the manuscript would deserve a deeper evaluation and discussion (e.g. from the density of ZSM-5 zeolite and the distribution of particle sizes determined from SEM, an estimate of the external specific surface area without the presence of mesopores can be calculated and compared with experimental data, etc.)
- Fig. 11 - FT-IR spectra of pyridine clearly show two distinguishable vibrational bands - 1454 and 1446 cm-1 in the region of Lewis acid centers. This fact is not discussed and explained in the manuscript, during the quantitative evaluation it is not clearly stated which band (or if both) was taken into account for the determination and which values of the molar absorption coefficients were used (or citation of the source of the coefficient values), which is important because it is possible to find significantly different coefficient values in the literature.
- a comparison of the catalytic performances of the prepared materials with "ordinary" ZSM-5 published in another study is not appropriate. First, this manuscript lacks any details about the "benchmark" material, and it is also not certain that the catalytic tests in ref. 30 were performed under identical conditions to those used in this study. It would be better to test the catalytic performance of "ordinary" ZSM-5 zeolite directly by the authors in the same catalytic apparatus under the same conditions.
From the above-mentioned reservations and suggestions for additions, it is clear that I can recommend the manuscript for publication only after a major revision.
Round 2
Reviewer 1 Report
"Point 2: Where is the error bar in Fig 14.
Response 2: Thank you for your careful observation. In the DTO conversion, we choose an on-line gas chromatograph (GC) equipped with flame detector (FID) to analyse the gas composition at the reaction outlet. After each gas collection, the chromatographic analysis time was 1 h, and the chromatographic won’t capture the outlet gas again in the process, resulting in only one set of data at each time point. The average and SD value cannot be calculated. Therefore, error bar was not marked in the catalytic evaluation results. In addition, this expression is very common in other similar articles. ( Microporous Mesoporous Mater., 2010, 132 384-394; Catal. Today, 2011, 171(1), 221-228; J. Catal., 2012, 290 (2012) 186-192; ACS Catal., 2016, 6, 11, 7311-7325; Appl. Catal., B, 2019, 243, 721-733) However, the reviewer's comments will be taken into consideration in future research."
You can repeat the analysis three time at least and you can calculate the error bar "to give meaning of the re-producibility" the physical meaning of error bar; that the re-producibility is it acceptable or not and in the same range or there is huge different. In addition, all references which you mentioned it in this comment not recently, check the newest. and what about reuse the catalyst for several runs!!!
Reviewer 2 Report
Authors aswered all my questions and revised the manuscritp accordingly.
Author Response
Thanks for your affirmation.
